# OpenReview forum: "SepVAE: a contrastive VAE to separate pathological patterns from healthy ones."
_ICML.cc/2023/Workshop/IMLH — IMLH 2023 Poster_

### Official Review · Reviewer_VCap · 2023-06-12
**A review for “SepVAE: a contrastive VAE to separate pathological patterns from healthy ones”**

**Rating:** 6
**Confidence:** 4

**Review:**

The paper adds two regularization terms, a disentangling term, and a classification term, to the CA-VAEs loss function, improving the existing CA-VAEs performance.

The paper considers an important problem in healthcare. The paper is easy to follow, and the proposed ideas are intuitive.

---

### Official Review · Reviewer_S9fp · 2023-06-16
**Clear and thoughtful**

**Rating:** 6
**Confidence:** 4

**Review:**

Pros:
1. This paper introduce two crucial regularization losses: a disentangling term between common and salient representations and a classification term between background and target samples in the salient space.
2. Develop a new Contrastive Analysis method: SepVAE.
3. Provide a fair comparison with other SOTA CA-VAE methods on 3 medical applications and a natural image (CelebA) experiment -> results show that the proposed approach outperforms previous CAVAEs methods.
4. Detailed ablation study to prove that the full model SepVAE is always better on average.

Cons:
1. some grammar errors -> improve writing
2. there are issues with inconsistent or incorrect capitalization throughout the paper

---

### Official Review · Reviewer_Fn2g · 2023-06-19
**Interesting idea but less impact in practice**

**Rating:** 4
**Confidence:** 4

**Review:**

This manuscript augmented the previous contrastive analysis VAE by introducing a disentangling term and a classification term to enhance the independence between common and salient spaces. The proposed SepVAE demonstrates some improvement in separating common and salient spaces compared to ConVAE and MM-VAE.

Despite the intriguing idea of generating disentangled common and salient features, this manuscript does not show how SepVAE can be impactful in real applications:

1. In the context of image editing and weakly supervised detection tasks (Fig. 1 and 5), SepVAE exhibits inferior performance on the CelebA dataset compared to GAN-based algorithms (e.g., [1]). SepVAE showed significant differences between the input and generated images.

2. Tables 1 to 4 employ latent vectors for classifying known attributes (e.g., glasses vs. no glasses or patient age). To evaluate the effectiveness of SepVAE, it would be more promising to compare its performance with that of a classifier trained directly on original images, measuring classification accuracy.

3. In the psychiatric disease experiment, the authors treat age, sex, and site as common variables assumed to be independent of psychiatric diseases. However, this claim may be problematic and requires further justification since many demographic factors are indeed strongly correlated with these diseases [2].

4. Please include the Silhouette score for Figure 6.

[1] Siddiquee, M.M.R., Zhou, Z., Tajbakhsh, N., Feng, R., Gotway, M.B., Bengio, Y. and Liang, J., 2019. Learning fixed points in generative adversarial networks: From image-to-image translation to disease detection and localization. In Proceedings of the IEEE/CVF international conference on computer vision (pp. 191-200).
[2] Gobinath, A.R., Choleris, E. and Galea, L.A., 2017. Sex, hormones, and genotype interact to influence psychiatric disease, treatment, and behavioral research. Journal of neuroscience research, 95(1-2), pp.50-64.

---

### Meta-Review · Area_Chair_odUq · 2023-06-20

**Recommendation:** Accept (Poster)
**Confidence:** 5

**Metareview:**

This paper proposes to use contrastive VAE to separate the common factors of variation between positive and negative samples. The paper has some merits to the medical imaging techniques. But there are several major concerns in addition to reviewers' comments. First, the use of contrastive learning for medical images in multiple modalities is not new, even with CA-VAE, separating patients and healthy subjects are well studied in the community. The paper lacks some fundamental literature studies. To this end, this paper can hardly prove its effectiveness and contribution to the peer works. As a long paper submission, the writing and flow are sufficient, and several experiments and evaluations are conducted. It's of the potential impact on evaluating VAEs for medical datasets.

---

### Decision · Program_Chairs · 2023-06-20

Accept (Poster)